Process of heavy metal transport between soil and the atmosphere: a review

Zhang Qiuye 1
Liu Hongyan hyliu@gzu.edu.cn 1 2 3
Li Xuexian 2
Liu Fang 2
1 College of Resources and Environmental Engineering, Guizhou University , Guiyang , Guizhou Province , China
2 College of Agriculture, Guizhou University , Guiyang , Guizhou Province , China
3 Key Laboratory of Karst Georesources and Environment, Ministry of Education, Guizhou University , Guiyang , Guizhou Province , China
Phairuang Worradorn
Electronic publication date: 2025 Dec 2
Publication date: 2025
Volume: 13
Electronic Location ID: e20381
Received 2025 Sep 3; Accepted 2025 Oct 22
Copyright: ©2025 Zhang et al.
Copyright year: 2025
Copyright holder: Zhang et al.
License: This is an open access article distributed under the terms of the Creative Commons Attribution License, which permits unrestricted use, distribution, reproduction and adaptation in any medium and for any purpose provided that it is properly attributed. For attribution, the original author(s), title, publication source (PeerJ) and either DOI or URL of the article must be cited.
License URL: https://creativecommons.org/licenses/by/4.0/

Keywords: Heavy metals, Transport process, Soil fugitive dust, Atmospheric deposition, Wind erosion

Funding: The National Natural Science Foundation of China (NSFC) No. 42067028 This work was supported by the National Natural Science Foundation of China (NSFC) (No. 42067028). The funders had no role in study design, data collection and analysis, decision to publish, or preparation of the manuscript.

==============================
The transport of heavy metals (HMs) (excluding Hg) between soil and the atmosphere significantly influences human production and life. This review systematically summarizes the processes involved in the wind erosion-driven transport of HMs from soil to the atmosphere and the partitioning of atmospheric HMs via atmospheric deposition, drawing on relevant literature analysis and synthesis. The results reveal that both soil and the atmosphere are sinks of HMs, influencing each other significantly. The transport of soil HMs to the atmosphere along with soil fugitive dust by wind force incorporates three pathways: the direct suspension of suspension-size aggregates, the collision and abrasion of creep-size and saltation-size aggregates, and the breakage or decomposition of creep-size aggregates. Conventional farming practices, elevated HM concentrations, and high wind speeds exacerbate soil HM emissions. However, the effects of soil organic matter (SOM) and clay on soil HM emission demonstrate dual characteristics. Atmospheric deposition has emerged as a significant source of soil HMs, with wet deposition predominating, except in arid and semiarid regions. Dry deposition is influenced by meteorological parameters and topographic profiles, whereas preceding weather and precipitation duration are other factors affecting for wet deposition. This process increases the exposure possibility and consequent exposure dosage of HMs to humans and crops, thereby amplifying the potential risks of HMs. Moreover, the capacity of atmospheric HMs for long-range wind-driven dispersal may leave remote and sensitive ecosystems that are increasingly vulnerable. Moreover, it concludes with a synthesis of the current challenges and discusses recommendations for future directions. Therefore, this review will have significant reference and guiding value for research in this field and is intended for researchers engaged in the migration of HMs in soil and atmosphere, the safe utilization of heavy metal contaminated soil, and regional background values of soil HMs.

Introduction

Soil and the atmosphere are intricately linked to human survival and production, exerting mutual influence. Soil fugitive dust, laden with a variety of pollutants, is dispersed into the atmosphere through wind forces (Feng, Sharratt & Wendling, 2011; Rubinstein, Ben-Hur & Katra, 2020; Sharratt, 2011), thereby contributing to atmospheric contamination. Inversely, soil can become contaminated through atmospheric deposition, whereby pollutants present in the atmosphere are settled to the ground surface, resulting in soil pollution (Eivazzadeh et al., 2021; Raja et al., 2015; Tang & Han, 2017; Tositti et al., 2020). Because HMs in soil and the atmosphere are deadly to production and life owing to their hazardous effects, bioavailability, and non-degradability (Fomba et al., 2018; Somayaji et al., 2022; Tchounwou et al., 2012), much attention has been paid to HMs. Although some trace quantities HMs are critical to life of human, animal,and plant, such as copper (Cu) and zinc (Zn), most HMs, such as mercury (Hg), cadmium (Cd), lead (Pb), arsenic (As), and chromium(Cr), have adverse effects on living organisms, even at low concentrations (Khan et al., 2008). Consequently, HMs have been recognized as one of the most critical constituents of global environmental pollutants that affect humans and biota (Ahmed et al., 2015; Carolin et al., 2017; Islam et al., 2014). For instance, Cd is linked to lung cancer, kidney dysfunction, nephrotoxicity and pulmonotoxicity, whereas Pb detrimentally affects the nervous, enzymatic, endocrine, and immune systems (Kotnala et al., 2025; Li et al., 2014; Jomova et al., 2024). HMs pose a threat to plant life and microorganisms, altering environmental physiological properties and, in extreme cases, killing cells, thereby reducing crop yield. Furthermore, the bioavailability of HMs can be concentrated in crops, which may threaten human health through the food chain (Imseng et al., 2018). Additionally, soil HMs enter the human body via particulate matter (PM) through inhalation and ingestion (Zhang et al., 2023; Fomba et al., 2018).

The transportation of HMs between soil and the atmosphere consists of two processes (1) the release of soil HMs into the atmosphere through soil fugitive dust (Feng, Sharratt & Wendling, 2011); and (2) the deposition of atmospheric HMs into the soil via atmospheric deposition (Feng et al., 2019a). Currently, industrial emissions, vehicular traffic, and agricultural activities are acknowledged as major anthropogenic sources owing to urbanization, industrialization, and the use of fertilizers and pesticides (Zhang et al., 2022). Both soil and the atmosphere serve as sinks for HMs, harboring multiple sources, thus complicating the identification of anthropogenic contributions to these two processes. In soil, HMs are absorbed onto soil colloids, including kaolin, ferrihydrite, humus, and certain minerals (Hashimoto & Yamaguchi, 2013; Karimian, Johnston & Burton, 2017; Wang et al., 2016). These components, which are integral to soil fugitive dust, can be readily mobilized to other places via wind forces, causing more widespread pollution. As to the water-driven transport of HMs, the contents of HMs decrease along the direction of water flow. Generally, HMs are absorbed by suspended particles in water and stored in sediments (Yang et al., 2024). Thereby, they can be released into the environment when hydrodynamic conditions change (e.g., due to dredging or erosion) or during flooding caused by extreme events such as heavy rainfall or tsunamis (Weber et al., 2018; Yang et al., 2024). The influence area of hydraulic migration is mainly located on both banks along the river (Ahmad et al., 2024; Egun et al., 2025; Liu et al., 2023a). Therefore, the transportation of wind force generates soil HMs distribution in a wider sphere than the transportation of water at the regional scale. In other words, hydraulic migration is even more difficult to predict for the common influence of topographic factors and climatic factors (Zhang et al., 2025a). Most atmospheric HMs adhere to PM, with only a fraction being associated with meteorological components (Feng et al., 2019a). Over time, atmospheric HMs are deposited onto surface soil through atmospheric deposition, contributing to their build-up on surface soil (Kelepertzis et al., 2020; Zgobicki et al., 2018). Furthermore, HMs within surface soil can be transported with surface runoff, accumulating in low-lying areas and water bodies, depending largely on the terrain and soil physicochemical properties (Likuku, 2006; Tositti et al., 2020). According to Chen et al. (2019) and Vithanage et al. (2022), inhalation, ingestion, and dermal contact are the primary HMs exposure pathways in particulate matter for human health, which increases the possibility of exposure and dosage for both crops and humans, posing a serious threat to agricultural productivity and human health. Numerous studies have focus on HM pollution in soil and the atmosphere, including their origin, spatial distribution, and ecosystem risks. However, the current literature offers limited comprehensive insights into the two processes for multiple impact factors, particularly the wind-mediated transport of soil HMs, which is critical for the pollution control of soil and the atmosphere. In this review, we outline the transport process of HMs from soil to the atmosphere by wind erosion and the partitioning process of atmospheric HMs on soil through atmospheric deposition, including the status and origin of HMs in soil and atmosphere, the two transport processes, and impact factors. Finally, the risks posed by HMs to ecosystems and human health during these processes are discussed. As Hg compounds are easily translated into elemental Hg (evaporation at room temperature) by microbes, Hg is excluded from this study.

Therefore, it is necessary to review and analyze the possible generation mechanism and influencing factors of this process, and summarize its possible impacts. This review will have significant reference and guiding value for research in this field and is intended for researchers engaged in the migration of HMs in soil and atmosphere, the safe utilization of HM contaminated soil, and regional background values of soil HMs, as well as offering references for environmental management (soil and atmosphere) and policy formulation by government department for contaminated soil by HMs (excluding Hg).

Survey Methodology

To identify relevant literature, a comprehensive search was conducted using databases such as China National Knowledge Infrastructure (CNKI) (https://www.cnki.net), Web of Science (http://webofscience.com), and Google Scholar. In this review, the topics of HMs, trace metals, cadmium, lead, fugitive dust, dust emission, wind erosion, wind tunnel, atmospheric deposition, dry deposition, wet deposition, source apportionment, source identification, and risk assessment from 1990 to 2025 were searched respectively. Then “heavy metal”, “heavy metals”, “trace metal”, “Cd”, and “Pb” were used to further selected these articles. Among them, more than 2,000 articles were in line with our theme. Furthermore, “soil”, “soils”, “atmosphere”, and “air” were utilized to screen above articles. Finally, 842 documents met the requirements, and through carefully reading and analysis, we selected the relevant articles suitable for this review. The selected references were classified by “soil”, “atmosphere”, and “cross-disciplinary fields”, which is beneficial for reading, understanding, and summarizing these references. Other references come from relevant books.

Conclusions

Transport process of HMs from soil to atmosphere

Soil is an important source of atmospheric HMs, particularly in highly polluted areas (Bi, Liang & Li, 2012; Gao et al., 2020; Zhang et al., 2022). Generally speaking, soil host is a large number of negatively charged particles, which makes it possible to adsorb positively charged metal ions. Additionally, soil constituents such as clay minerals and oxides might physically and chemically adsorb HMs (Chen, 2018). Studies that specifically target the HM emission process from soil to the atmosphere are still very few, and because of the underlying processes, there is still a lot of unreported data, which is likely related to environmental science and wind erosion. Based on previous studies regarding fine particle emissions from soil into the atmosphere, it is reasonable to hypothesize that the transport process of HMs may be affected by factors such as HM concentration, wind speed, and various physical and chemical properties (e.g., soil organic matter (SOM), soil moisture content, and particle/aggregate size) (Feng, Sharratt & Wendling, 2011; Li et al., 2015; Tatarko et al., 2020; Tatarko et al., 2021).

Origination of HMs in soil

There has been a lot of discussion about the problem of soil pollution caused by HMs in recent years. The sources of HMs in soil are mainly natural (rock weathering and pedogenesis), industrial, agricultural, traffic-related, and atmospheric deposition (Cloquet et al., 2006; Huang et al., 2019; Wang et al., 2021; Wang et al., 2019a; Wang et al., 2019b; Wu et al., 2020; Zhuo et al., 2019a; Zhang et al., 2025b). Natural sources connected to stratigraphy and lithology typically contribute less to the concentrations of HMs in soil. However, in areas with a high geological background, such as Guizhou, Guangxi, and Yunnan provinces in China (Chen et al., 2015), the total concentrations of HMs can remain high under conditions of minimal anthropogenic influence, thereby posing a potential threat (Zhang et al., 2022). Metal smelting is widely used in several industries (Cloquet et al., 2006; Kang et al., 2021; Peng et al., 2022; Yu et al., 2022), including mining (Cloquet et al., 2006), the phosphorus chemical industry (Jiao et al., 2012), thermal power plants (Raja et al., 2015), and the electronics industries (Yang et al., 2019). These enterprises can emit a large amount of soot containing a lot of HMs and produce many solid wastes through fuel combustion, equipment wear, and sewage treatment that contaminate the surrounding environment (Deng et al., 2024; Zhang et al., 2025b). Agricultural sources largely include fertilizers, pesticides, green manure, and livestock manure (Zhang et al., 2022). Traffic sources are predominantly associated with the combustion of gasoline or diesel, tyre wear, brakes, three-way catalytic converters, and road surfaces (Taghvaee et al., 2018; Gérardin & Midoux, 2016; Wang et al., 2019b). Atmospheric deposition includes these sources, which will be further discussed in the upcoming sections.

Transport process

The transport of HMs from soil to the atmosphere is important for the spatial distribution of soil HMs, as the introduction of soil HMs into the atmosphere occurs concurrently with the emission of soil fugitive dust. According to Kolesar et al. (2022), wind and the surface are involved in the driving force of this process, which promotes the movement and deposition of sand and dust by the earth and makes it easier for sand to migrate. Some studies have confirmed that large aggregates possess high cohesion force, leading to their are inability to move or blow up (Feng, Sharratt & Wendling, 2011; Li et al., 2015; Kolesar et al., 2022). Hence, soil aggregate size is significant for resistance stability. Based on Hagen’s classification, soil aggregates are categorized into three sizes: creep-size aggregates (0.84–2.00 mm), saltation-size aggregates (0.10−0.84 mm), and suspension-size aggregates (<0.1 mm) (Feng, Sharratt & Wendling, 2011; Hagen, Wagner & Skidmore, 1999). Building on this, three theories have been proposed to elucidate the emission of soil fugitive dust (Gelbart & Katra, 2020; Kolesar et al., 2022) (Fig. 1): (1) wind force blows up suspension-size aggregates; (2) wind force causes saltation and creep of creep-size aggregates and saltation-size aggregates on the surface, subsequently colliding with and abrading other aggregates, producing soil fugitive dust; and (3) creep-size aggregates generate soil fugitive dust as a result of breakage or decomposition when they creep on the Earth’s surface. Currently, wind tunnel experiment is widely used to simulate this process (Feng, Sharratt & Wendling, 2011; Li et al., 2015; Liang et al., 2024).

Figure 1 Theories regarding the emission of soil particulate matter and their associated heavy metals (HMs).

Impact factors and mechanism

Wind speed.

Based on the transport process discussed in section “Transport process”, the emission of soil fugitive dust requires wind power to impart an initial thrust to soil aggregates, thereby inducing their suspension, saltation, and creep (Lackóová, Kaletová & Halászová, 2023; Lin et al., 2024). Wind speed serves as an index for measuring the wind power. Some studies have shown a positive correlation between the emission fluxes of soil particles and wind speed at the same sampling height (Feng, Sharratt & Wendling, 2011; Li et al., 2015). This correlation likely accentuates the loss of soil HMs, implying that HM emission fluxes increase in tandem with those of soil particles. At lower wind speeds, the emission fluxes of particles are minimal. However, as the wind speed increases, so do the soil aggregates, which in turn cause saltation and creep; as a result, smaller aggregates are released into the air, which raises the emission threshold (Rubinstein, Ben-Hur & Katra, 2020). Nonetheless, the emission of soil particles is also influenced by other factors, thereby significantly complicating the quantitative analysis of wind force.

Concentration of HMs in soil.

The concentration of soil HMs is an important factor for atmospheric HMs, as soil HMs can be transported into the atmosphere in the form of soil fugitive dust. Several studies have found that the higher the concentration of HMs in soil, the more likely these metals are to be emitted into the atmosphere (Li et al., 2016b; Liu et al., 2022; Wu et al., 2020; Yu et al., 2022). Moreover, particles originating from anthropogenic sources tend to be smaller, accumulate easily in surface soil, and are more readily mobilized into the atmosphere by aerodynamic forces. At present, there are no direct studies demonstrating the influence of HMs concentration on this process; hence, further research of the relevant mechanisms is necessary.

Soil water.

Soil water, comprised of adsorbed and capillary water, can effectively reduces soil fugitive dust by enhancing the cohesive force of soil particles, leading to a reduction in HMs loss from soil. Many studies have shown that soil particle loss from soil is inversely proportionate to soil water content, with soil erodibility gradually decreasing towards zero as the soil water concentration increases (Baker, Southard & Mitchell, 2005; Funk et al., 2008; Gill, Zobeck & Stout, 2006; Madden, Southard & Mitchell, 2009). This decrease is due to the weakened cohesive forces between soil particles resulting from the loss of adsorbed and capillary water, along with the increased mass of soil particles (Madden, Southard & Mitchell, 2010). However, some studies have demonstrated that this phenomenon is associated with the transition of soil water from capillary to adsorbed water (Cornelis & Gabriels, 2003; Madden, Southard & Mitchell, 2010). Consequently, both adsorbed and capillary water are pivotal in reinforcing the binding force of soil particles, with capillary water contributing more substantially (Madden, Southard & Mitchell, 2010). There exists a threshold moisture level, roughly one-third of the water content retained by soil at −1.5 MPa (soil water potential), beyond which soil erodibility initially decreases gradually with increasing water content, followed by a rapid decline with each additional increment of water, ultimately reaching zero (Madden, Southard & Mitchell, 2008). Even if adsorbed water is an important binding force in the soil, especially in clayey soil (materials) and the total amount that makes sand, and in the case of clay, the capillary water is not enough to support cohesion in sand and clay (Madden, Southard & Mitchell, 2010).

Soil organic matter (SOM).

Soil organic matter (SOM) is primarily from plant and animal residues, which also has an effect on the quantity and stability of big aggregates, which is a major factor in determining the resistance of the soil to erosion (Li et al., 2015). With the accumulation of SOM, both cation exchange capacity (CEC) and the number of large aggregates increase incrementally (Aimar et al., 2012; An & Xu, 1988). In addition, SOM can effectively enhance microbial diversity (Chen, 2018) and reduce soil emissions. Li et al. (2015) found that PM loss was not significantly related to soil SOM across three wind speeds (8, 10, and 13 m/s) for crushed soil, whereas there was a significant negative correlation at lower speeds for uncrushed soil. Additionally, several studies have shown a positive relationship between SOM content and HM concentration in soil (Bastakoti, Robertson & Alfaro, 2018; Kabata-Pendias, 2017), indicating that SOM effectively immobilizes HMs. Thus, the role of SOM in the loss of soil HMs is dual-faceted (promoting and inhibiting) (Fig. 2) and requires further exploration in future studies.

Figure 2 Effect of soil organic matter (SOM) on loss of heavy metals (HMs) in soil.

Soil aggregate size.

The size of soil aggregates is utilized to evaluate the soil’s erodibility (Sirjani et al., 2024). The Wind Erosion Prediction System (WEPS) categorizes soil loss size classes into three groups: saltation and creep (ranging from 0.1 to 2.0 mm), suspension (smaller than 0.1 mm), and smaller than 0.01 mm. These classifications are useful for evaluating the environmental air quality impacts of wind erosion (Feng, Sharratt & Wendling, 2011; Tatarko & Wagner, 2007). Based on this, a more detailed partition, as introduced in section “Transport process”, was subsequently proposed. Both the percentage of aggregates with <0.84 mm diameter and the geometric mean diameter (GMD) are pivotal parameters for evaluating soil susceptibility to wind erosion (Hagen, Wagner & Skidmore, 1999). Li et al. (2015) indicated that the GMD and percentage of aggregates with <0.84 and <0.42 mm diameter bear no significant relationship with the loss of PM10 (particulate matter with diameters below 10 µm in soil fugitive dust) and PM2.5 (particulate matter with diameters below 2.5 µm in soil fugitive dust) for crushed soil, potentially due to man-made alteration of the original aggregate composition (Fig. 3A). With respect to uncrushed soil, the proportion of aggregates with <0.84 and <0.42 mm diameter correlates positively with the loss of PM10 and PM2.5, but these correlations were not statistically significant (p > 0.05) (Fig. 3B). The GMD of uncrushed soil (natural soil) was inversely correlated with the loss of PM10 and PM2.5, but this correlation was not significant (Fig. 3B).

Figure 3 Correlation of loss of PM10 and PM2.5 with soil aggregate size for crushed soils (A) and uncrushed soils (B) at three wind speeds (8, 10, and 13 m/s).

The data deriving from Li et al. (2015).

Soil particle size.

Soil particle size (dispersed size) signifies the primary particle composition of soil, while aggregate size (nondispersed size) is employed to evaluate the state of soil particles in situ (Feng, Sharratt & Wendling, 2011). The potential of soil to produce PM10 or PM2.5 has been gauged by the content of ≤10 µm and ≤2.5 µm in soil. By reviewing the related literature, the correlations between the content of ≤10 µm and ≤2.5 µm, and the ≤10μm/≤2.5 µm ratio with the emission of PM10 and PM2.5 were established. The conclusions are presented in Fig. 4. The contents of ≤10 µm and ≤2.5 µm in soil are negatively associated with the loss of PM10 and PM2.5. The relationship was significantly negative for all experiments, with PM10 loss more readily influenced by the contents of ≤10 µm and ≤2.5 µm. Moreover, the uncrushed soil at low wind velocities were more susceptible to the impact of ≤10 µm and ≤2.5 µm. However, the correlation between 2.5 µm/10 µm and PM loss varied noticeably with the wind velocity. Moreover, the ratio of 2.5 µm/10 µm exhibited a negative correlation in Fig. 4A for crushed soil. Moreover, in the case of uncrushed soil (Fig. 4B), there was no significant association seen between the 2.5 µm/10 µm ratio and PM loss.

Figure 4 Correlation of loss of PM10 and PM2.5 with dispersed size of crushed soils (A) and uncrushed soils (B) at three wind speeds (8, 10, and 13 m/s).

The data deriving from Li et al. (2015).

Soil texture refers to the mechanical composition of the soil, which, based on international classification, is divided into sand (0.02–2 mm), silt (0.0002–0.02 mm), and clay (<0.0002 mm). Figure 5 shows the correlation of PM loss with sand, silt, and clay in the soil. The summary reveals a negative correlation between clay content and the loss of PM10 and PM2.5 for both crushed soil and uncrushed soil at all three speeds, especially notable for PM10 loss. In contrast, silt and sand were positively correlated with the loss of PM10 and PM2.5 for crushed soil. For uncrushed soil, silt and sand were positively correlated with PM10 loss but demonstrated no significant correlation with PM2.5 loss. Thus, anthropogenic activities can markedly increase the potential for PM loss. The main components of sand are silicon dioxide and various other minerals, resulting in sand particles that show minimal to no cohesiveness, making them easily dislodged and carried by the wind (Li et al., 2015; Rubinstein, Ben-Hur & Katra, 2020). Gelbart & Katra (2020) affirmed that there was a negative and positive correlation between the amount of sand and clay and PM loss. Considering that the primary constituents of clay are secondary clay minerals, which are distinguished by their high specific surface areas and stable cation exchange properties (Ben-Hur et al., 2009; Eden et al., 2020; Miguel Reichert, Darrell Norton & Favaretto, 2001), clay exhibits exceptional cohesive characteristics (Zuo et al., 2024). It forms wind-resistant clods that provide greater resistance to wind erosion and contribute more wind-resistant materials for abrasion compared to sand and silt (Haynes & Swift, 1990). This, in turn, helps to reduce the transportation of HMs from soil into the atmosphere. Although clay improves stability against wind erosion, it can also absorb a considerable amount of HMs (Proust, Fontaine & Dauger, 2013; Sipos et al., 2021), leading to increased HM loss. Therefore, clay exerts a dual effect, both promoting and inhibiting the loss of HMs in soil (Fig. 6).

Figure 5 Correlation of loss of PM10 and PM2.5 with texture of crushed soils (A) and uncrushed soils (B) at three wind speeds (8, 10, and 13 m/s).

The data deriving from Li et al. (2015).

Thus, soil particle and aggregate sizes influence the emission of soil fugitive dust, which, in turn, indirectly impacts the emission of soil HMs. The HM content in different particle sizes varies, serving as a critical determinant of HM loss from soil. Several studies have discovered that the concentration of HMs in particle rises as particle size decreases (Bi, Liang & Li, 2012; Shao et al., 2022). This is attributed to smaller particles having a higher specific surface area,which amplifies the presence of iron and manganese oxides, sulfides, organic matte, and clay minerals. Consequently, the HMs were adsorbed more easily onto these fine particles. Given that small particles are more susceptible to wind-driven atmospheric entry, the emission fluxes of HMs are also expected to increase accordingly. Nevertheless, as no studies have yet reported on the mechanisms of HMs transportation from soil to the atmosphere, the impact of soil particles and aggregate size necessitates further investigation.

Farming practices.

Land use patterns can reflect the effects of anthropogenic activities on soil owing to differences in physicochemical properties (Baker, Southard & Mitchell, 2005; Eden et al., 2020; Funk et al., 2008). Among the various land-use types, agricultural land is more susceptible to tillage (Gantulga et al., 2023; Yu et al., 2022; Wang et al., 2022). Conventional tillage alters the mechanical composition of soil, leading to the disintegration of soil aggregates and production of smaller and looser particles prone to saltation (Swet & Katra, 2016; Katra, 2020). This enhances the emission of soil fugitive dust, subsequently increasing the release of HMs from soil (Funk et al., 2008; Goossens, Gross & Spaan, 2001; Sharratt, 2011; Li et al., 2015). Moreover, a number of different farming methods, including those related to farming and cropping, also result in different dust emissions (Baker, Southard & Mitchell, 2005; Madden, Southard & Mitchell, 2008; Madden, Southard & Mitchell, 2009; Sharratt & Schillinger, 2014), thereby influencing the emission of soil HMs. In addition, the application of agricultural inputs, such as straw, soil conditioners, soil amendments, fertilizers, green manure, and livestock manure, can alter the size of soil aggregates and soil texture, influencing the emission of soil particles (Li, Bair & Parikh, 2018). Zhang et al. (2022) determines concentrations of HMs in fertilizer and livestock manure, and their concentration of Cd are 0.12 and 8.31 mg/kg, accounting for 0.19% and 21.15% of Cd in soil, respectively. Therefore, these agricultural inputs also introduce considerable quantities of HMs into soil, thereby increasing the possibility of soil HMs being released into the atmosphere.

Transport process of HMs from the atmosphere to soil

HMs in atmosphere

The atmosphere is one of the major HM carriers, primarily through a suspended mixture of solid and liquid particles (Vithanage et al., 2022). PM-bound HMs are mainly enriched in atmospheric particles within a size range of 0.1–0.3 µm (Connan et al., 2013). The coarse particles that are classified as atmospheric particles (PM10-aerodynamic diameter smaller than 10 µm) and the fine particles (PM2.5-aerodynamic diameters smaller than 2.5 µm) are the general categories of atmospheric particles, as well as the ultrafine particles (UFPs) (aerodynamic diameter smaller than 0.1 µm) (Pope III & Dockery, 2006). Coarse particles are predominantly generated from volcanoes, farming, mining, roads, sea spray, windstorms, and deserts, and primarily originate from combustion processes, such as those from gasoline or diesel combustion, wood burning, mining, thermal power plants, and other industries (e.g., cement, melters, steelworks, paper mills, etc.) (Kumar et al., 2020; Omidvarborna, Baawain & Al-Mamun, 2018). UFPs are produced during fossil fuel combustion and the condensation of semi-volatile substances (Pope III & Dockery, 2006). Additionally, PM-bound HMs can undergo long-range transport with airborne particles, especially fine particles, significantly extending their reach significantly (Liu et al., 2018b; Sun et al., 2020; Yadav et al., 2019).

Figure 6 Effect of clay on loss of heavy metals (HMs) in soil.

Atmospheric HMs commonly originate from both natural and anthropogenic sources. Natural sources include soil, volcanic dust, and volcanic gases (Azimi et al., 2005). Industrial activities, mining, fossil fuel combustion, traffic-related emissions, and incineration of urban solid waste are widely regarded as the main contributors of airborne HMs (Li et al., 2016b; Liang et al., 2019; Mohanraj, Azeez & Priscilla, 2004; Shao et al., 2022; Yu et al., 2022). Moreover, HMs in the atmosphere surrounding roads are often influenced by the traffic volume, type of vehicular traffic, road age, speed limits, and the industrial nature of the area (Bernardino et al., 2019; Bernardino et al., 2021; De Silva et al., 2021). The use of anthropogenic resources has gradually grown to be a significant source of atmospheric HMs,particularly in industrial settings. For instance, Kang et al. (2021) revealed that there were between 1,200, 35,000, and 20,900 mg/kg of Cd, Pb and Zn in soil fugitive dust surrounding a zinc smelter, respectively; and Liu et al. (2022) also found that the concentration of Cd in the atmospheric deposition of the Wanshan Hg mine area is 1.21 mg/kg, significantly exceeding the background levels of local surface soil. Additionally, the concentration of HMs in the atmosphere is influenced by environmental factors and meteorological conditions, such as seasonal variation, wind direction, wind speed, and temperature (Cheng et al., 2014; Hovmand et al., 2008).

Transport process and impact factors

Atmospheric HMs significantly contribute to soil HMs through atmospheric deposition, which is defined as the process of atmospheric pollutants entering terrestrial and aquatic environments, encompassing wet and dry deposition (Connan et al., 2013; Gunawardena et al., 2013; Han et al., 2014). Atmospheric deposition is a scavenging process for the atmosphere, whereas it is a pollution process for soil (Kara et al., 2014; Yun, Yi & Kim, 2002).

Dry deposition.

Dry deposition is the direct deposition of atmospheric particles without rainfall or snowfall (Feng et al., 2019a). The variation in particle size directly influences the duration of the atmospheric residence (Deshmukh, Deb & Mkoma, 2012), subsequently leading to a discrepancy in dry deposition rates and, correspondingly, dry deposition flux. It is important to note that fine particles exhibit a lower dry deposition rate compared to coarse particles (Fang et al., 2004; Sakata & Asakura, 2011). This deposition phenomenon is of particular concern in urban and industrialized areas, which not only have higher emissions but also lack effective scavenging factors such as vegetation (Weerasundara et al., 2017). Numerous studies on dry deposition worldwide have been reported and summarized by Vithanage et al. (2022) and Wu et al. (2018). According to several studies, traffic activity has been found to be a key factor in the accumulation of atmospheric HMs in urban areas (Tasdemir et al., 2006; Weerasundara et al., 2017; Werkenthin, Kluge & Wessolek, 2014). Furthermore, long-range transport of particles is another origin of dry deposition, with particle size playing a determinant role. Considerable dust production from the Taklimakan Desert, the Loess Plateau, and the Gobi Desert transports both natural and anthropogenic HMs along migration pathways, which have a significant negative effect on the water bodies and the soil of East Asia (Li et al., 2016a). For example, there have been reports of dust storms in Korea that were linked to the movement of particles from the Gobi Desert (Alashan semi-desert), the Loess Plateau, and large industrial areas of China (Han et al., 2004). Inevitably, dry deposition is also influenced by meteorological parameters such as wind direction, seasonality, and topographic profiles (Tositti et al., 2020; Negral et al., 2021; Zhong et al., 2021). Because of the interception effect of mountains, Pb in atmospheric particles can enter alpine systems during long-range transport (Bing et al., 2016; Zhang et al., 2013). Zhong et al. (2021) and Zhang et al. (2025a) also showed that the deposition of atmospheric Pb in alpine forest is modulated by terrain. Low altitudes are susceptible to local anthropogenic sources, whereas high altitudes are more prone to be influenced by long-range transport (Zhang et al., 2022).

Wet deposition.

Wet deposition, another pathway of atmospheric deposition, represents a process where atmospheric particles, scavenged within and below cloud formations, descend to the ground surface during rainfall or snowfall events after dissolution in clouds and adsorption onto droplets (Azimi et al., 2005; Feng et al., 2019a). Generally, the concentrations of HMs in wet deposition correlate positively with their solubility, which depends on the pH of rainwater and the origin of the HMs. Studies have indicated that crustal elements such as Al, Fe, Si, Co, and Mn in wet deposition are less soluble, whereas anthropogenic elements (e.g., Cd, Pb, Zn, Cu, and Ni) show higher solubility (Vithanage et al., 2022). Consequently, the concentration of anthropogenic HMs in wet deposits usually exceeds that of naturally sourced HMs (Vithanage et al., 2022; Wu et al., 2018). Moreover, the concentration of HMs in wet deposition is influenced not only by the preceding weather but also by the duration of precipitation. Muezzinoglu & Cizmecioglu (2006) and Weerasundara et al. (2017) suggested that an upward trend in the concentration of HMs during wet deposition when rainfall followed an extended dry period. Interestingly, a significant amount of HMs is washed away during the initial phase of rainfall (first few minutes), resulting in higher concentrations of HMs in wet deposition over short durations compared to longer durations (Vithanage et al., 2022).

Superposition of atmospheric HMs to soil

The superposition of atmospheric HMs on soil via atmospheric deposition has become increasingly prominent in recent years, especially in developing countries (Peng et al., 2019). For instance, the atmospheric deposition flux of Cd is 4.0 g/ha/yr (0.4–25 g/ha/yr) in China, which is much higher than that in Europe (0.4 g/ha/yr) (Six & Smolders, 2014). Approximately 35% of Cd in surface soil of China is attributed to atmospheric deposition (Luo et al., 2009), with this proportion exceeding 40% in mining areas (Liu et al., 2022). This contribution rate can reach 50%–93% for other elements (As, Cr, Hg, and Pb) (Feng et al., 2019a). Generally, the input of wet deposition to the ground surface exceeds that of dry deposition (Wu et al., 2018). Based on the summary and calculation, an estimated 68%–74% of Cd and Zn originate from wet deposition, while 25%–33% come from dry deposition (Vithanage et al., 2022). Additionally, the distribution of Pb in soil at various elevations is regulated by wet deposition (Zhang et al., 2025a; Zhang et al., 2025b; Zhong et al., 2021). In contrast, dry deposition is the primary contributor to HMs in atmospheric deposition in arid and semi-arid regions (Liu et al., 2019). Regional differences in climate, pollution sources, and background values have led to variations in HM settlements. Table 1 illustrates the deposition fluxes of HMs worldwide. Remote and background areas, which are less influenced by anthropogenic activities, exhibited relatively lower fluxes of HMs than urban and metropolitan areas. Owing to urbanization, the atmospheric deposition of HMs in some general cities exceeds that in some metropolises. Atmospheric deposition also influences the spatial distribution of HMs in soil, as wind forces disperse soil HMs into the atmosphere, depositing them elsewhere, and thus widening their distribution. As a result, historical HMs are difficult to identify easily from natural sources over time. In some historical industrial areas, despite the closure or relocation of enterprises, HMs persist in the soil and form a potential source of HMs (Peng et al., 2022; Zhang et al., 2022). Atmospheric deposition not only increases the pollution of HMs but also alters the mineral phase composition of the soil, thereby enhancing the soil’s adsorption capacity for HMs and its bioavailability (Liu et al., 2023b). Nevertheless, HMs in atmospheric deposition can influence soil enzymes and microbial community structure and function, thereby reflecting the impact of anthropogenic pollution (Raja et al., 2015). Moreover, HMs can undergo physiochemical reactions in the atmosphere, altering their properties and greatly significantly affecting their mobility in soil following deposition. Chrastný et al. (2011) indicated that the mobility of atmospheric HMs depends on their retention time in the atmosphere, and the longer they linger, the more stable the HMs become. As a result, more research is necessary because of the substantial effect of atmospheric deposition on soil HMs.

Table 1 Atmospheric bulk deposition of heavy metals (HMs) in different locations of the world (g/ha/yr).

Location	Heavy metal (HMs)	Remarks	Monitoring time	Reference	
	Cd	Hg	Pb	Cr	As	Cu	Ni	Zn				
Yangtze River Delta, China	4.1	–	359.0	132.0	15.7	139.0	46.0	895.0	Urban cluster	2006–2007	Huang et al. (2009)	
Pearl River Delta, China	0.7	–	127.0	64.3	–	186.0	83.5	1,040.0	Urban cluster	2001–2002	Wong et al. (2003)	
Beijing, China	2.4	–	111.3	72.90	52.6	58.7	34.7	330.8	Metropolis	2007–2008	Guo, Lyu & Yang (2017)	
0.8	–	1.7	0.2	29.3	–	–	–	1984	
Guangzhou, China	3.1	–	–	71.4	21.6	191.0	76.7	1,020.0	Metropolis	2010–2011	Huang et al. (2014)	
Xi’an, China	–	–	180.0	–	26.0	77.0	44.0	510.0	Metropolis	2007–2008	Cao et al. (2011)	
Tokyo, Japan	3.9	–	99.0	62.0	29.0	160.0	68.0	–	Metropolis, Harbor	2001–2002	Sakata, Tani & Takagi (2008)	
Mumbai, India	4.5	–	24.0	2.7	–	5.0	119.0	653.0	Metropolis	2001	Gajghate, Pipalatkar & Khaparde (2012)	
Los Angeles, USA	–	–	69.0	16.8	–	76.7	18.9	438.0	Metropolis	2002–2003	Lim et al. (2006)	
Xiangtan, China	11.4	–	156.5	37.8	58.9	64.1	15.7	730.9	General city	2016–2018	Feng et al. (2019a)	
Zhuzhou, China	21.3	–	223.0	–	–	–	–	–	General city	2017–2018	Feng et al. (2019b)	
Daejeon, South Korea	0.2	–	17.7	10.3	3.8	45.2	5.2	56.5	General city	2007	Lee, Choi & Kang (2015)	
Changji, China	5.8	–	228.5	–	–	80.6	44.7	–	General city	2016	Liu et al. (2019)	
Daya Bay, China	1.6	–	76.3	29.5	36.4	46.7	19.1	939.6	Industrialized city, Harbor	2015–2017	Wu et al. (2018)	
Baoding, China	9.8	–	458.0	–	–	289.0	66.6	1,221.0	Industrialized city	2015–2017	Pan & Wang (2015)	
Varna, Bulgaria	0.11	–	6.31	9.12	–	16.17	4.38	229.22	Harbor	2008–2009	Theodosi et al. (2013)	
Eastern Adriatic, Mediterranean region	0.11	–	5.1	–	–	15.4	12.8	49.5	Harbor	2019–2020	Penezić et al. (2021)	
Zunyi, China	1.168	–	–	–	–	–	–	–	Background area	2020–2021	Cui et al. (2022)	
Marais Vemier, France	1.1	1.1	17.0	–	–	–	43.4	357.6	Background area	2010–2012	Connan et al. (2013)	
Matsuura, Japan	2.9	–	76.9	46.2	12.8	42.1	44.9	–	Remote area	2004–2006	Sakata & Asakura (2011)	
Balearic Islands, Spain	–	–	2.0	–	–	3.0	2.0	19.0	Remote area	2010–2012	Cerro et al. (2020)	

Ecosystem and human health risk of HMs at the transport process

Ecosystem risk

The transport of HMs between soil and the atmosphere increase the potential for increased exposure to humans and plants. Plants are capable of assimilating soil HMs through their root system (Chen et al., 2021), a process involving transporters and ion diffusion (Wei et al., 2016). When soil HMs are transferred from soil to the atmosphere by wind, the concentration of atmospheric HMs correspondingly increases. Concurrently, plant foliage has a heightened likelihood of contacting with HMs, thus bolstering the assimilation of HMs via foliar uptake (Przybysz, Nersisyan & Gawroński, 2018; Shahid et al., 2017). Atmospheric HMs infiltrate plants through assimilation of leaf epidermis and stomata (Deng et al., 2024; Feng et al., 2019b; Li et al., 2022), and are then disseminated to other tissues (Ma et al., 2019). Some studies have confirmed that atmospheric deposition is the main cause of Pb in lettuce, dry soil and cabbage, especially in polluted areas (Gao et al., 2021; Schreck et al., 2014). Therefore, the mobilization of soil fugitive dust facilitates the entry of more soil HMs into the food chain, thereby further amplifying the ecosystem risk of HMs.

Furthermore, atmospheric HMs enter soil through atmospheric deposition. Generally, the concentration of HMs in atmospheric deposition exceeds that in soil because soil particles suspended by wind force carry a high content of HMs, and anthropogenic HMs are perpetually discharged into the atmosphere by various pollution sources (Kang et al., 2021; Luo et al., 2019; Zhang et al., 2022). Consequently, the concentration of soil HMs gradually increases over time owing to the cumulative effect of atmospheric deposition (Yu et al., 2022; Zhang et al., 2022). In addition, atmospheric HMs are characterized by high mobility and bioactivity, especially under the influence of anthropogenic sources, which is helpful for the migration of HMs from soil to plants and poses a direct threat to human health through the food chain. Moreover, the long-range transport of PM-bound HMs suggests that HMs originating from anthropogenic sources such as coal combustion and vehicular emissions can contaminate remote regions via atmospheric deposition (Siudek & Frankowski, 2017; Zang et al., 2021).

Furthermore, given the different toxicities shown by different HMs, risk assessment methods were used to evaluate the specific ecosystem effects and risks associated with individual HMs, rather than focusing only on concentration levels. For instance, the potential ecological risk index (RI) has been adopted to evaluate the potential ecological risk presented by HMs (Kang et al., 2020). However, the toxic response factor for each HM differs, leading to variations in the potential ecological risk (Zhuo et al., 2019b). Numerous studies have investigated the effects of specific HMs (Cd, Pb, Cr, Cu, Ni, and Zn) on ecosystems (Peng et al., 2018; Taati et al., 2020; Wang et al., 2019b). Among them, Cd represents a more severe risk owing to its high toxicity and the proportion of exchangeable and carbonate fractions compared to other elements (Luo et al., 2019; Sun et al., 2010).

Human health risk

The transport of particles laden with HMs between soil and the atmosphere enables HMs to enter the atmospheric cycle, posing significant health risks (Liu et al., 2018a; Rajkumar et al., 2025). Atmospheric PM is associated with respiratory and cardiovascular diseases (Galvão et al., 2020; Huang et al., 2019; Samet et al., 2000). Concurrently, HMs can cause serious damage to the renal, neurological, gastrointestinal, reproductive, cardiovascular, and hematological systems (Li et al., 2014). Regarding the health risks associated with HMs in atmospheric particles and their deposition, three primary exposure pathways affect human health ingestion, inhalation, and dermal contact (Deepak et al., 2024; Okechukwu et al., 2022; Peng et al., 2017). Ingestion and inhalation are regarded as the primary exposure pathways for HMs in atmospheric dust (Lu et al., 2014). Therefore, the transport of HMs between soil and the atmosphere increases the possibility of direct contact and exposure to HMs, further amplifying the human health risk of HMs. U.S. EPA (2019) employs a methodology that uses the Hazard Quotient (HQ) to assess the risk associated with each HM. The Hazard Index (HI) and Carcinogenic Risk (CR) provide measures of the total non-carcinogenic and carcinogenic risks of each element through the three exposure routes. For non-carcinogenic risks, HQ and HI values >1.0 suggest that HM pollution may be harmful to the human body, while values <1.0 suggest minimal to no harm (Lai et al., 2024; Zhou et al., 2024). For CR, values <1 ×10−6 denote no carcinogenic risk, and values of 1 ×10−6–1 ×10−4 suggest acceptable carcinogenic risk, and values >1 ×10−4 indicate unacceptable risk (Chen et al., 2015; Hu et al., 2012; Singh et al., 2025). Moreover, some studies have shown that children are more sensitive to HMs because of their higher body surface area to volume (Chen et al., 2015; Lu et al., 2014; Mugoša et al., 2016). Generally, the human health risks posed by HMs originating from natural sources are relatively low. However, the impact of anthropogenic sources significantly amplifies the risk associated with HMs, particularly carcinogenic risk (Chen et al., 2019; Ali-Taleshi, Feiznia & Masiol, 2022).

Current challenge and future direction

The transport of HMs between soil and the atmosphere is a complex process that requires further research and understanding. Although numerous studies have investigated the migration process of soil fugitive dust from land surfaces to the atmosphere, additional research is required to develop and refine related theories and explanations. Furthermore, the existing mechanisms for the loss of PM may not be directly applicable to the emission of HMs, emphasizing the need for further investigation. In addition, the precision of trace metal detection, such as Cd, Ni, Cu, in simulation experiments should be improved. Both soil and atmosphere are sinks of HMs, and understanding the relative impacts of natural sources and various anthropogenic sources is also need to be further researched. Thus, it is essential to gain a thorough understanding of the entire migration process of HMs between soil and the atmosphere, as well as to evaluate the effects of anthropogenic HMs on soil and atmospheric conditions.

The next sections may help with the study that has to be done in the following ways:

(1) Further exploration of the impact factors and mechanisms influencing the production of soil fugitive dust, based on previous studies by Feng, Sharratt & Wendling (2011), Madden, Southard & Mitchell (2010), Li et al. (2015), and Tatarko et al. (2020); Tatarko et al. (2021), in order to further improve the understanding of this process.

(2) Investigation of the transport process of HMs from soil to the atmosphere, considering the physical and chemical characteristics of soil (such as soil particle size, aggregate size, SOM, and HM concentration) and assessing whether existing mechanisms for these factors are applicable to the emission of HMs.

(3) Proposing research on the complete “soil-atmosphere-soil” migration process of HMs, which would enable the refined calculation of the transport coefficient of every element in the transport process, the identification of HMs originating from anthropogenic and natural sources, and provide insights into the superposition of atmospheric HMs in soil.

(4) Further quantitative evaluation of the potential risks of HMs in the process based on the findings.

Addressing these research directions will contribute to a more comprehensive understanding of the process and facilitate the improvement of risk assessment and management strategies in the future.

Figure 7 Potential influence of heavy metals (HMs) transport process between soil and the atmosphere.

Conclusions

This study provides a comprehensive overview of the transport process, including the entry of soil-derived HMs into the atmosphere via dust dispersion and the deposition of atmospheric HMs back into soil through atmospheric deposition. The status and origination of HMs in soil and the atmosphere, transport processes, and impact factors re discussed. The risk of HMs to ecosystems and human health from these processes is also discussed. This process and its impact on ecosystems and human health are summarized in Fig. 7, and the key conclusions of the study are as follows:

(1) Soil and the atmosphere are both sinks for HMs, encompassing natural and anthropogenic sources, and they interact significantly with each other.

(2) Soil HMs enter into atmosphere along with soil fugitive dust carried by wind. This process involves three pathways: the direct suspension of suspension-size aggregates, the collision and abrasion of creep-size and saltation-size aggregates, and the breakage or decomposition of creep-size aggregates.

(3) The emission of soil HMs is influenced by the concentration of soil HMs, wind speed, tillage practices, and physical and chemical properties (e.g., SOM, soil water, and particle/aggregate size). Conventional tillage, high HMs concentrations, and high wind speeds promote the loss of HMs. The effects of SOM and clay on the loss of soil HMs demonstrated dual characteristics, with both promotion and inhibition observed.

(4) When it comes to deposition of the atmosphere, which includes both wet and dry deposition, it has become a major source of soil HMs, with wet deposition being the main cause of the problem,except in arid or semi-arid areas.

(5) Dry deposition is influenced by meteorological parameters and topographic profiles, whereas preceding weather and precipitation duration are other important impact factors for wet deposition.

(6) Soil fugitive dust and atmospheric deposition increase the exposure potential and dosage of HMs for humans and crops, posing risks to ecosystems and human health. Atmospheric HMs are able to travel long-range distances with the help of wind force, potentially causing sensitive ecosystems in remote areas.

Therefore, some measures, such as increasing of vegetation coverage and conserving of soil and water, should be taken to reduce HMs transportation between soil and the atmosphere. Further research is required to better the knowledge on this process and develop more effective mitigation methods.

Additional Information and Declarations

Competing Interests

Author Contributions

Data Availability

The authors declare there are no competing interests.

Qiuye Zhang conceived and designed the experiments, performed the experiments, analyzed the data, prepared figures and/or tables, authored or reviewed drafts of the article, and approved the final draft.

Hongyan Liu conceived and designed the experiments, authored or reviewed drafts of the article, and approved the final draft.

Xuexian Li performed the experiments, authored or reviewed drafts of the article, and approved the final draft.

Fang Liu performed the experiments, analyzed the data, prepared figures and/or tables, and approved the final draft.

The following information was supplied regarding data availability:

This is a literature review.

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
