# Peer review of "Process of heavy metal transport between soil and the atmosphere: a review"

_PeerJ, doi:10.7717/peerj.20381_

## Round 0.1 · original submission · Minor Revisions

· Academic Editor

Minor Revisions

Upon reviewing the references, it is evident that all except one, dated 2025, originate from 2022 or before. This indicates that articles from recent years have been disregarded. This post is a review, and such a situation is unacceptable.

**Language Note:** The review process has identified that the English language must be improved. PeerJ can provide language editing services - please contact us at [email protected] for pricing (be sure to provide your manuscript number and title). Alternatively, you should make your own arrangements to improve the language quality and provide details in your response letter. – PeerJ Staff

·

Basic reporting

The subject of the study is truly excellent. I can say that I was quite excited when I started reading the article. However, upon deeper investigation, I saw that the article needed extensive revision. First of all, if you examine the references, you will see that all references except for one from 2025 are from 2022 and earlier. This shows that publications from recent years have not been taken into consideration. However, this article is a review article, and such a situation is unacceptable. My other suggestions are listed below.

Experimental design

This study is a review study. I did not understand why WOS and CNKI were used in the literature review. It would have been more logical to use Scopus or Google Scholar instead of WOS because these databases are more widely accepted internationally. In addition, using only WOS would also have been a more logical decision. Furthermore, biomonitors should have been included in the keywords. There is a wealth of information on this topic in studies of this kind. Many publications I am aware of and that are relevant to the topic have not been evaluated in this article. Numerous studies conducted in regions such as Europe and America have been overlooked.

Validity of the findings

Due to the shortcomings I have outlined above, there are numerous deficiencies in the findings section. For example, in section Line 135, human activities in urban areas—such as fuel combustion, equipment use, human waste, etc.—have not been evaluated as sources of heavy metals. Furthermore, mining activities, which are among the most significant sources of heavy metals, have not been mentioned at all. Furthermore, the phytoremediation effect of plants on heavy metal concentrations in soil and air has been overlooked. However, the effect of heavy metal pollution on plants has been mentioned. Numerous studies have addressed the effective use of plants in monitoring and reducing changes in heavy metal pollution. Moreover, when heavy metals are mentioned, only long-known heavy metals such as Pb, Cr, and Ni are discussed. However, recent studies have focused on heavy metals such as As, Ba, Tl, Sr, Se, and V, which are dangerous to human health even at very low concentrations. Moreover, these heavy metals are included in the priority pollutant list by organizations such as the EPA and ATSDR. The fact that these heavy metals are not mentioned in this study is also due to the failure to review recent studies.

Additional comments

The topic of the study is truly excellent and addresses a much-needed area of research. Although the article has reached a certain point, it is unacceptable in its current state. I would like to congratulate the authors on their choice of topic and the process that has brought them to this stage. Once the article has been developed, I am sure it will be used by many researchers and will receive a high number of citations. However, it needs significant development. After the specified corrections and additions have been made, I will be able to review the article in much greater detail.

Reviewer 2 ·

Basic reporting

no comment

Experimental design

no comment

Validity of the findings

no comment

Additional comments

The manuscript addresses an important and timely topic—the bidirectional transport of heavy metals between soil and atmosphere—with clear relevance to environmental science and human health. The literature base is comprehensive and the research objective is well defined. However, several issues should be addressed to strengthen the manuscript before it can be considered for publication. Below I provide my detailed comments.
1. Language clarity and readability
The manuscript contains numerous long and complex sentences, which hinder readability. For instance, the sentence “When it comes to soil, atmospheric pollutants are contaminated by atmospheric deposition...” is confusing and should be rephrased for clarity. A thorough language revision is recommended to make the text more concise and academic.
2. Conceptual definitions
Terms such as “soil HMs,” “atmospheric HMs,” and “fugitive dust” are used frequently but not consistently defined. Providing clear and consistent definitions at the outset would improve accessibility for interdisciplinary readers.
3. Logical structure of the Introduction
While the Introduction covers heavy metal hazards in detail, this section appears imbalanced. The discussion of toxic effects (e.g., Cd, Pb) is lengthy, while the central theme—the transport processes between soil and atmosphere—is underdeveloped. The narrative should be reorganized to emphasize mechanisms and knowledge gaps more clearly.
4. Mechanistic depth
Critical aspects such as the dual role of soil organic matter (SOM) and clay in heavy metal retention and release, as well as the distinction between wind-driven and water-driven transport, are mentioned but not sufficiently explained. Expanding these points would make the review more insightful.
5. Figures
The clarity of the values in Figures 3 and 4 should be improved; some labels and numbers are not clearly visible.
6. References
The manuscript cites a wide range of studies, indicating a solid foundation of existing research. However, the number of references published in the past three years is relatively small. Incorporating more recent literature would strengthen the currency of the review.

---

## Round 0.2 · accepted · Accept

· Academic Editor

Accept

This revised version is suitable for publication in PeerJ.

·

Basic reporting

The revision was deemed sufficient. The article can be accepted as is.

Experimental design

The revision was deemed sufficient. The article can be accepted as is.

Validity of the findings

The revision was deemed sufficient. The article can be accepted as is.

Additional comments

The revision was deemed sufficient. The article can be accepted as is.